# New Mutants of Epsilon Toxin from *Clostridium perfringens* with an Altered Receptor-Binding Site and Cell-Type Specificity

**DOI:** 10.3390/toxins14040288

**Published:** 2022-04-16

**Authors:** Jonatan Dorca-Arévalo, Inmaculada Gómez de Aranda, Juan Blasi

**Affiliations:** 1Department of Pathology and Experimental Therapeutics, Faculty of Medicine and Health Sciences, Campus of Bellvitge, University of Barcelona, Hospitalet de Llobregat, 08907 Barcelona, Spain; igomezdearanda@ub.edu (I.G.d.A.); blasi@ub.edu (J.B.); 2Biomedical Research Institute of Bellvitge (IDIBELL), L’Hospitalet de Llobregat, 08908 Barcelona, Spain; 3Institute of Neurosciences, University of Barcelona, 08035 Barcelona, Spain

**Keywords:** epsilon toxin, *Clostridium perfringens*, pulpy kidney disease, lectin, pronase E, N-glycosidase F, β-elimination, proximal tubules, MDCK cells, enterotoxemia

## Abstract

Epsilon toxin (Etx) from *Clostridium perfringens* is the third most potent toxin after the botulinum and tetanus toxins. Etx is the main agent of enterotoxemia in ruminants and is produced by *Clostridium perfringens* toxinotypes B and D, causing great economic losses. Etx selectively binds to target cells, oligomerizes and inserts into the plasma membrane, and forms pores. A series of mutants have been previously generated to understand the cellular and molecular mechanisms of the toxin and to obtain valid molecular tools for effective vaccination protocols. Here, two new non-toxic Etx mutants were generated by selective deletions in the binding (Etx-ΔS188-F196) or insertion (Etx-ΔV108-F135) domains of the toxin. As expected, our results showed that Etx-ΔS188-F196 did not exhibit the usual Etx binding pattern but surprisingly recognized specifically an O-glycoprotein present in the proximal tubules of the kidneys in a wide range of animals, including ruminants. Although diminished, Etx-ΔV108-F135 maintained the capacity for binding and even oligomerization, indicating that the mutation particularly affected the pore-forming ability of the toxin.

## 1. Introduction

Epsilon toxin (Etx) is the third most potent clostridial toxin after the botulinum and tetanus toxins. It is produced by *Clostridium perfringens* types B and D and causes severe and rapidly fatal enterotoxemia in ruminants, causing high mortality and economic losses [1,2,3]. Etx is produced as a very low-active molecule, the epsilon prototoxin (pEtx), which becomes fully active after proteolytic cleavage of the C- and N-terminal residues by trypsin and chymotrypsin in the intestinal lumen of the host or by the lambda protease produced by *C. perfringens* [4,5]. Etx alters the intestinal barrier [6], enters the gut vasculature, and permeabilizes vascular endothelia [7], leading to various histological changes that include vascular endothelial damage and generalized edema in the brain, lungs, and heart [8]. Intravenous (i.v.) administration of radiolabeled Etx in mice has shown an accumulation of Etx in the brain and kidneys, as well as small amounts in the heart, lungs, liver, and stomach [9]. Etx also has the capacity to cross the blood–brain barrier (BBB) [10,11], which has been linked to Etx toxicity [12]. Moreover, incubations of brain slices with Etx have shown specific binding of the toxin to myelinated structures [13]. In addition, it has been suggested that Etx could play a role in multiple sclerosis [14], causing the death of mature oligodendrocytes and central nervous system demyelination [15,16]. The receptor for Etx is still under debate. Three proteins have been suggested as candidates for being the Etx receptor, including hepatitis A virus cellular receptor 1 (HAVCR1), the myelin and lymphocyte protein MAL, and caveolin-1 (CAV1) [15,17,18,19,20]. MAL appears to be the most likely candidate, as it is the only protein that has been demonstrated to be necessary for both Etx binding and cytotoxicity. However, this remains controversial [19,20,21].

A few cell lines are sensitive to Etx, the most commonly studied being the Madin-Darby canine kidney (MDCK) cell line with a renal origin [22]. In MDCK cells, Etx has a high-affinity single binding site with a dissociation constant (K_d_) of around 4 nM [23]. More recently, new cell lines sensitive to Etx have been described, such as Fischer rat thyroid (FRT) cells with a thyroid origin [24], the human acute lymphoblastic leukemia T cell line MOLT4 [20], and the endothelial 1G11 cell line derived from mouse lungs [25]. In all cases, the mode of action of Etx follows a three-step scheme that is similar in all cell types: binding of Etx to the receptor on the plasma membrane, oligomerization and insertion, and the formation of a pore that leads to ion deregulation and adenosine triphosphate (ATP) depletion in the cytoplasm, subsequently leading to cell death [20,22,24,26].

In accordance with the three-step mechanism of action, the crystal structure of Etx has three domains that share a similar structural conformation to those of other bacterial pore-forming toxins (PFTs) of the aerolysin-like family, such as aerolysin from *Aeromonas hydrophila* and other β-pore-forming bacterial toxins [27]. In fact, the pore-complex structure of Etx has recently been defined by cryo-electron microscopy, providing new insights into the molecular details of the structural rearrangements during pore formation [28]. Domain 1 has the only tryptophan present in the molecule and is implicated in receptor binding. Moreover, domain 1 has a cluster of aromatic residues (Tyr29, Tyr30, Tyr36, Tyr42, Tyr43, Tyr49, Tyr196, Tyr209, and Phe212), a feature observed in other sugar-binding proteins [17,27]. Indeed, defective binding of Etx together with an impaired ability to kill host cells has been observed in mutants in which the aromatic rings present in this domain have been mutated, with the tyrosine residue replaced by glutamic acid [17]. Domain 2 is an amphipathic region with a long loop flanked by serine- and threonine-rich sequences that could have a role in insertion, protein oligomerization, and pore formation [27,29]. Two dominant-negative mutants of Etx (I51C/A114C and V56C/F118C), involving mutations in this domain, have been developed in which site-specific mutations have been designed to include cysteine substitutions to form a disulfide bond that prevents the unfolding of the loop involved in Etx insertion into the plasma membrane. One cysteine in each mutant is included in the membrane-insertion domain of the toxin and another cysteine is included in the protein backbone, which impairs oligomerization and cytotoxicity in the host cell in vitro [30]. Moreover, the mutant Etx-H106P [31] has one histidine substituted by a proline residue in the loop, which results in a non-toxic form. Curiously, fusing these mutants with the green fluorescent protein (GFP) revealed that the mutant GFP-Etx-I51C/A114C was toxic in MDCK cells at a higher concentration dose, although the mutants GFP-Etx-V56C/F118C and GFP-Etx-H106P were not toxic under the same conditions. However, the ability to oligomerize was conserved, suggesting the possibility of dissociating the oligomerization step from the subsequent pore-forming activity [12]. Lastly, domain 3, which is similar to domain IV of aerolysin, is also implicated in membrane insertion. It contains the carboxyl-terminal portion of the toxin that is associated with heptamerization and oligomerization [27]. The carboxyl-terminal peptides appear to block oligomerization. When these peptides are not removed during the activation of the prototoxin, the disrupted interaction between the monomers subsequently blocks oligomerization [27,29].

Several Etx mutants have been generated to understand the cellular and molecular mechanisms of the toxin and to obtain non-toxic molecules as candidates for efficient vaccinations [1,31,32]. These mutants present deficient or no activity in binding, oligomerization, and pore formation. In all cases, these mutants have been shown to be useful as molecular tools. Here, two new mutants were generated: Etx-ΔV108-F135, in which the insertion loop from domain 2 was deleted, and Etx-ΔS188-F196, in which the beta-sheet located in domain 1, implicated in the Etx binding site, was deleted. We observed a strongly decreased binding of Etx-ΔV108-F135 and a completely abolished binding of Etx-ΔS188-F196 to its receptor. Moreover, both mutants were not toxic both in vitro and in vivo, even at the high concentrations tested. Curiously, Etx-ΔS188-F196 showed a new binding site in the epithelium of the proximal tubules of the kidneys in all the species tested.

## 2. Results

### 2.1. Design and Production of the Mutants

To develop new non-toxic mutants of Etx, the sequence that encodes the loop in domain 1 and another that encodes the amphipathic loop in domain 2 were deleted, and two new mutants were generated: Etx-ΔS188-F196 and Etx-ΔV108-F135, respectively (Appendix A).

We produced the mutants as a fusion protein with the GFP. The inactive and active form of the toxin was named pEtx and Etx, respectively, as shown in Appendix A.

### 2.2. In Vivo Studies 

#### In Vivo Studies: Histological Analysis of Mice Injected with GFP-Etx Mutants

Previous studies using fluorescence microscopy to analyze GFP-Etx-injected mice have demonstrated specific binding of Etx to blood vessels in the brain and the distal tubules of the kidneys, resulting in the death of these epithelial cells [33,34]. Moreover, Etx crosses the BBB [10], and a direct correlation has also been described between the cytotoxic effect of the toxin in the kidneys and the ability to cross the BBB [12]. Following a similar procedure, we studied the effect of the newly designed mutants in mice. Injected GFP-Etx-ΔV108-F135 and GFP-Etx-ΔS188-F196 did not show any evidence of toxicity or lethality, at least up to 2 h post-injection compared to GFP-Etx. GFP-Etx-injected mice died between 5 and 10 min after the injection. Accordingly, mice injected with GFP-Etx-H106P, GFP-Etx-ΔV108-F135, or GFP-Etx-ΔS188-F196 were also processed 10 min after the injection.

Histological analysis of the kidneys from GFP-Etx-injected mice showed pyknotic nuclei (condensed chromatin) in the epithelial cells of the distal kidney tubules (arrowheads, Figure 1D), similar to previous observations in Etx-intoxicated animals [12,34]. 

This demonstrated a cytotoxic effect of GFP-Etx on these cells. Conversely, no alterations were observed in the kidneys of animals injected with GFP-Etx-H106P, GFP-Etx-ΔV108-F135, or GFP-Etx-ΔS188-F196 (arrows, Figure 1A–C, respectively). 

In addition, immunohistochemical assays of brain samples revealed that the GFP-Etx-H106P, GFP-Etx-ΔV108-F135, and GFP-Etx-ΔS188-F196 mutants were not able to cross the BBB and that GFP-Etx-ΔS188-F196 did not bind to the vascular endothelium (arrowhead, Appendix A). GFP-Etx-H106P, GFP-Etx-ΔV108-F135, and GFP-Etx were able to bind to the endothelium (arrows, Appendix A) of the brain microvessels. Only GFP-Etx crossed the BBB, obtaining access to the brain parenchyma (asterisk, Appendix A).

### 2.3. In Vitro Studies

#### 2.3.1. In Vitro Studies in MDCK Cells

##### Study of the Cytotoxicity of GFP-Etx Mutants and ATP Depletion in MDCK Cells

To study the cytotoxicity of the mutants, MTS assays were performed in MDCK cells. The active GFP-Etx-ΔV108-F135 and GFP-Etx-ΔS188-F196 forms were compared to the toxic GFP-Etx as a positive control and to GFP-pEtx as a negative control. The results showed that GFP-Etx-ΔV108-F135 and GFP-Etx-ΔS188-F196 were not toxic to MDCK cells (Figure 2A), as around 100% of the cells were alive after 30 min of exposure.

The pore-forming capacity of Etx can be quantified by measuring the amount of ATP released from the cytosol or internal cell stores [20,24,25]. To quantify the amount of ATP released, luciferin-luciferase assays were performed. MDCK cells were incubated with GFP-Etx-ΔV108-F135 (blue line, Figure 2B) or GFP-Etx-ΔS188-F196 (purple line, Figure 2B), and their effects were compared to those of the non-toxic GFP-Etx-H106P (green line, Figure 2B) and GFP-pEtx (black line, Figure 2B) used as negative controls, as well as to those of GFP-Etx (red line, Figure 2B) used as the positive control. No ATP release was observed with any of the GFP-Etx mutants. As expected, all the ATP was released following GFP-Etx exposure and no residual ATP was measured after cell permeabilization with Triton X-100 at the end of the experiment (insert, red line, Figure 2B). However, all the ATP content was detected in the cells treated with GFP-pEtx, GFP-Etx-H106P, GFP-Etx-ΔV108-F135, or GFP-Etx-ΔS188-F196 after cell permeabilization with Triton X-100 (insert, black, green, blue and purple lines, respectively, Figure 2B). 

To show that the lack of activity of the new mutants in MDCK cells was not due to the addition of the bulky GFP molecule, MTS assays were performed, which corroborated that both the GFP- and non-GFP-tagged versions of the Etx mutants were not toxic to MDCK cells (Appendix A).

##### Binding and Oligomerization of GFP-Etx Mutants in MDCK Cells 

GFP-pEtx is a useful tool to study the binding properties of Etx [10,13,34] together with the non-toxic mutant GFP-pEtx-H106P, which also shows the same binding pattern as wild-type Etx [12]. To study the binding of the new mutants, MDCK cells were incubated with GFP-pEtx-ΔV108-F135 or GFP-pEtx-ΔS188-F196 and compared to those incubated with GFP-pEtx. While GFP-pEtx-ΔV108-F135 showed a clearly reduced binding to the MDCK plasma membrane compared with GFP-pEtx (Figure 3D–F), GFP-pEtx-ΔS188-F196 did not show any binding to the MDCK plasma membrane (Figure 3G–I). 

There was also no binding of GFP alone (used as a negative control) to the MDCK plasma membrane (Figure 3J–L).

Western blot analysis showed the formation of an oligomer complex with a high molecular weight in MDCK cells treated with the positive control GFP-Etx (lane 2, Figure 4).

However, a thinner band was observed in MDCK cells treated with the non-toxic GFP-Etx-H106P (lane 3, Figure 4), and a fainter band was detected in the cells treated with GFP-Etx-ΔV108-F135 (lane 4, Figure 4). No oligomer complex formation was detected in the cells treated with GFP-Etx-ΔS188-F196 (lane 5, Figure 4) nor in those incubated with GFP-pEtx (negative control; lane 1, Figure 4).

#### 2.3.2. Binding of GFP-Etx Mutants in the Kidneys

GFP-pEtx binds in vitro to the distal and collecting tubules of the kidneys in mice and ruminants [23,34]. To study the binding of the new mutants in the kidneys, we incubated mouse, sheep, cow, and goat slides with the GFP-pEtx mutants, comparing the binding with that of GFP-pEtx used as a positive control. Distal tubules were stained by GFP-pEtx in all the species tested (arrows in Appendix A), while a diminished binding was detected for GFP-pEtx-ΔV108-F135 (arrows in Appendix A). 

Curiously, although the distal tubules were not stained by GFP-pEtx-ΔS188-F196, this mutant showed selective binding to the proximal tubules in all the species tested (arrowhead in Appendix A). 

To confirm that the binding of GFP-pEtx-ΔS188-F196 was restricted to the proximal tubules, sections of mouse kidney were co-incubated with the GFP-pEtx mutant and the lectins: PNA, which recognizes the distal tubules [35], and DSA, which recognizes the proximal tubules [36,37]. GFP-pEtx-ΔS188-F196 and the DSA lectin showed colocalization in the proximal tubules (arrows, Figure 5), but no colocalization was detected between the PNA lectin and GFP-pEtx-ΔS188-F196 (Figure 5D–F).

Co-incubations of GFP-pEtx-ΔS188-F196 with the unlabeled pEtx-ΔS188-F196 in excess showed a decrease in GFP-pEtx-ΔS188-F196 binding to the proximal tubules, demonstrating that it was a specific binding (Appendix A).

Taking advantage of this result and in order to stain the proximal and distal tubules at the same time, the Etx mutants pEtx-H106P and pEtx-ΔS188-F196 were labeled with DyLight Fluor 488 and 550, respectively, as explained in the Materials and Methods section. In all the species tested, pEtx-H106P-DyLight488 recognized the distal tubules (green, Figure 6A,D,G,J), while pEtx-ΔS188-F196-DyLight550 recognized the proximal tubules (red, Figure 6B,E,H,K).

As pEtx also binds to the urothelium in vitro [23], we compared the binding of the positive control GFP-pEtx with that of the GFP-pEtx mutants to bladder sections from sheep and goat. Binding was detected in the urothelium for GFP-pEtx-ΔV108-F135 in all the species tested (Appendix A), but it was less intense compared to that of GFP-pEtx. 

No binding was detected in the urothelium for GFP-pEtx-ΔS188-F196 (Appendix A), similar to that observed in the distal and collecting tubules of the kidneys and MDCK cells.

To further characterize the nature of the binding of GFP-pEtx-ΔS188-F196 in the kidneys, we performed a series of experiments using pronase E, detergents, N-glycosidase F, and β-elimination, similar to a previous study characterizing the binding of GFP-pEtx [23].

Mouse kidney sections were incubated with pronase E (Figure 7) to enzymatically remove a wide range of proteins that could be involved in the binding of GFP-pEtx-ΔS188. In this case, the binding of GFP-pEtx-ΔS188-F196 to the epithelial cells from the proximal tubules of the kidneys was clearly decreased compared to the control untreated kidney slice (arrows in Figure 7D,F).

To study the implication of a lipid environment in the binding of GFP-pEtx-ΔS188-F196, some treatments with detergents (sodium cholate, sodium deoxycholate, and Triton X-100) at 2% and at room temperature (RT) were performed with the kidney slides. There was no decrease in the GFP-pEtx-ΔS188-F196 binding in any of the cases, even there was also a slight increase in the binding to the proximal tubules compared to the control (arrows, Figure 8).

To check the implications of the glycans in the binding of pEtx-ΔS188-F196 to the epithelial cells of the proximal tubules, several treatments with N-glycosidase F (to hydrolyze the N-glycan chains from glycoconjugates) and β-elimination (to hydrolyze the O-glycan chains from glycoconjugates) were performed. After N-glycosidase F treatment, the binding of GFP-pEtx-ΔS188-F196 and PNA did not change compared to the control (compare Figure 9K,L with Figure 9H,I), but the binding of DSA was significantly reduced after enzymatic action (compare Figure 9E,F with Figure 9B,C).

β-elimination treatment for 1 h abolished the binding of GFP-pEtx-ΔS188-F196 to the proximal tubules (compare Figure 10A,G with Figure 10D,J). Consequently, the binding of the PNA lectin, which recognizes O-glycans, was abolished (compare Figure 10K,L with Figure 10H,I), while the binding of the DSA lectin, used for N-glycan detection, was increased (compare Figure 10E,F with Figure 10B,C). 

## 3. Discussion

Etx binds to and damages the epithelial cells from the distal and collecting tubules of mouse kidneys [12,33,34]. In fact, post-mortem findings in sheep infected with *C. perfringens* type D show dark and jelly-like kidneys that are attributed to the effect of Etx [38]. Previous studies have demonstrated that the recombinant protein GFP-pEtx is a convenient tool to study the binding mechanism of Etx since it binds to the same sites as the active toxin [10,13,34]. 

Here, we present two new Etx mutants: Etx-ΔV108-F135, in which the loop responsible for the insertion into the plasma membrane has been deleted, and Etx-ΔS188-F196, in which the loop in domain 1 involved in receptor binding has been deleted. Moreover, we also produced their respective fluorescent forms linked to GFP. In vivo experiments with these new mutants injected into mice revealed that they were non-lethal, did not cross the BBB, and were not neurotoxic. The distal and collecting tubules of the kidneys were preserved after the injection of Etx mutants compared to the pyknotic nuclei present in the distal and collecting tubules of mice injected with wild-type Etx. In addition, Etx-ΔV108-F135 could bind to the endothelium of the brain microvasculature, but could not cross into the brain parenchyma like wild-type Etx could. Etx-ΔS188-F196 did not bind to the endothelium of the brain microvasculature, which was attributed to the deletion of the loop present in domain 1 that is directly involved in the binding of Etx to its receptor.

In vitro experiments using MDCK cells revealed that the mutants did not have a cytotoxic effect and did not trigger the release of ATP from the cells, indicating that they were not inserted into the plasma membrane or did not form the pore complex. However, Etx-ΔV108-F135 could bind to the plasma membrane. Nonetheless, this binding was highly reduced compared to that of wild-type Etx, suggesting that the integrity of the structure is required for adequate high-affinity binding. Etx-ΔS188-F196 did not bind to the plasma membrane, which was attributed to the deleted loop in domain 1 that is mainly involved in the binding of Etx to its receptor. Accordingly, if domain 1 is not conserved, the binding to the receptor on target cells is disrupted, and if domain 1 is conserved, the binding to the receptor can be modulated by domain 2. Western blots showed that the mutant GFP-Etx-H106P was able to oligomerize although the oligomer was not inserted into the plasma membrane and did not produce pores or cause cell death. Furthermore, in the Western blot, GFP-Etx-ΔV108-F135 showed a very faint band corresponding to less oligomerization of this Etx mutant, which was in accordance with the reduced binding and absence of pore formation observed in the MDCK cells. GFP-Etx-ΔS188-F196 did not oligomerize in the plasma membrane, which was consistent with its inability to bind to a specific receptor, a step that precedes oligomerization. 

Incubations of kidney and bladder sections from different species, including the natural targets of Etx, with the Etx mutants produced results similar to those observed in the MDCK cells, as GFP-pEtx-ΔV108-F135 was able to bind to the distal and collecting tubules in the kidneys and to the urothelium in the bladder, although the binding was highly reduced compared to that of wild-type Etx. Curiously, GFP-pEtx-ΔS188-F196 did not bind to the distal and collecting tubules of the kidneys or to the bladder urothelium. However, it bound to the proximal tubules of the kidneys in all the species tested. This was supported by the colocalization of GFP-pEtx-ΔS188-F196 with the lectin DSA, which recognizes the proximal tubules. In fact, co-incubations with pEtx-ΔS188-F196-DyLight550 and pEtx-H106P-DyLight488 also revealed distinct binding to the proximal and distal tubules, respectively. Furthermore, this binding was specific as co-incubation of GFP-pEtx-ΔS188-F196 with an excess of the unlabeled pEtx-ΔS188-F196 markedly reduced the binding of GFP-pEtx-ΔS188-F196 to the proximal tubules. We speculate that the mutation in domain 1 in Etx-ΔS188-F196 could also produce a change in the conformation of domain 2 and/or 3, which would not only impair binding to the receptor (and moved to another binding site) but also toxicity, as the in vivo experiments showed. This mutant was completely non-toxic although it recognized the kidney proximal tubules without producing any toxic effects in them, as shown in the histopathological analysis of the kidneys from mice injected with GFP-Etx-ΔS188-F196. In any case, more experiments should be carried out, including in vitro assays in epithelial cell lines with a proximal tubule origin and from different species to further characterize the involvement of this Etx mutant in the recognition of a new binding site in the kidneys.

To characterize the binding of GFP-pEtx-ΔS188-F196 to the kidney proximal tubules, we performed different treatments in kidney tissue sections incubated with GFP-pEtx-ΔS188-F196. Treatments with pronase E reduced the binding of GFP-pEtx-ΔS188-F196 to the proximal tubules, suggesting that the receptor has a protein moiety. On the contrary, treatments with different detergents were not able to reduce the binding of GFP-pEtx-ΔS188-F196 to the proximal tubules. Moreover, the binding of GFP-pEtx-ΔS188-F196 after these treatments was increased in all cases. This effect of the detergents increasing receptor availability has been shown in several models [39,40]. Basically, detergents make the receptor more accessible to the ligand compared to normal conditions [40,41]. According to our results, lipids were not involved in the binding of GFP-pEtx-ΔS188-F196, since its binding was not affected by high detergent concentrations or the use of aggressive detergents such as the non-ionic detergent Triton X-100 with very low critical micelle concentrations. Treatment with N-glycosidase F did not induce any modification in the binding of GFP-pEtx-ΔS188-F196, although it somewhat increased it when compared to wild-type Etx. This could be explained by some N-linked glycans partially covering or unmasking the binding site for Etx [23]. On the contrary, β-elimination treatment abolished the binding of GFP-pEtx-ΔS188-F196. All these findings suggested that the binding of GFP-pEtx-ΔS188-F196 is mediated through an O-glycosylated protein and that lipids are not involved. 

Moreover, we speculate that these new mutants, Etx-ΔV108-F135 and Etx-ΔS188-F196, could be a useful tool to generate new vaccines against Etx in order to study the immunogenicity and the production of neutralizing antibodies although further studies should be performed.

## 4. Conclusions

We can conclude that the loop present in domain 1 is directly involved in the binding of Etx to target cells, while the loop present in domain 2 is required for the binding in addition to oligomerization and pore formation, since Etx binding and subsequent oligomerization are greatly reduced by the deletion of the loop from domain 2. The Etx mutant lacking the loop from domain 1 showed changes in its binding pattern compared to that of wild-type Etx. This mutant specifically recognized the proximal tubules of the kidneys in a wide range of species, including the natural hosts for *Clostridium perfringens* type B and D infections. The binding of the new mutant seemed to be mediated by an O-glycoprotein, and lipids were not involved. In addition, mutations in domain 2 altered the binding of Etx to its receptor. These results deserve further studies and could be used to further characterize the toxin-receptor interactions.

## 5. Materials and Methods

### 5.1. Cloning, Expression, and Purification of pEtx and pEtx Mutants with or without GFP and Their Respective Active Forms

In all cases, wild-type and mutant DNA was cloned into the pGEX-4T-1 vector (Amersham Biosciences; Freiburg, Germany) with or without the EGFP coding sequence to produce the corresponding recombinant fusion protein, as previously described [10,42].

pEtx was produced starting at amino acid A47, therefore lacking the N-terminal peptide but not the C-terminal peptide. Mutations were introduced using the QuikChange Multi Site-Directed Mutagenesis kit (Stratagene, La Jolla, CA). pEtx-H106P, with or without GFP, was generated as previously described [12,31]. pEtx-ΔV108-F135 and pEtx-ΔS188-F196, with or without GFP, were generated following the manufacturer’s instructions and using the primers listed in Table 1. The nomenclature for the mutants follows that previously used by [31]. All active forms of Etx and the Etx mutants, with or without GFP, were generated using the reverse active Etx primer (Table 1).

All plasmids were transformed into the RosettaTM (DE3) pLysS Escherichia coli strain for optimal protein expression. Briefly, protein expression was induced overnight in the presence of 1 mM isopropyl beta-D-thiogalactopyranoside (IPTG) at RT in 250 mL of LB medium containing 50 µg/mL of ampicillin. Cells were pelleted and resuspended in 20 mM ice-cold phosphate buffer (PB), pH 7.5, with 250 mM NaCl, before being sonicated and centrifuged at 12,000× *g* for 20 min at 4 °C. The resulting supernatant was incubated with Glutathione SepharoseTM 4B beads (GE Healthcare Life Sciences) for 1 h at 4 °C. Finally, the recombinant protein was eluted by thrombin cleavage in PB containing 2.5 mM CaCl_2_.

To analyze the recombinant protein, 10 µL of each sample were electrophoresed in a precast polyacrylamide gel (Mini-PROTEAN^®^ TGX ^TM^; #456-9033, Bio-Rad, Gaithersburg, USA). The gel was then transferred to a nitrocellulose membrane (Trans-Blot^®^ Turbo TM; #1704158, Bio-Rad) and analyzed by Western blot using the anti-GFP rabbit polyclonal antibody (1:1000 dilution [23]) or the anti-Etx rabbit polyclonal antibody (1:1000 dilution [34]), followed by polyclonal swine anti-rabbit immunoglobulins/HRP (1:5000 dilution, #P0217, Dako, Carpinteria, USA). The membrane was developed with LuminataTM Crescendo Western HRP substrate (Millipore, Billerica, USA), and signals were detected using an Amersham Imager 600 (GE Healthcare Life Sciences). 

### 5.2. Production of pEtx-H106P-DyLight488 and pEtx-ΔS188-F196-DyLight550

pEtx-H106P-DyLight488 and pEtx-ΔS188-F196-DyLight550 were labeled using the DyLightTM 488 (#46402, Thermo Fisher Scientific, Waltham, USA) and 550 (#62262, Thermo Fisher Scientific) NHS Ester Kit, respectively, according to the manufacturer’s instructions.

Briefly, the NHS ester-activated dye labels proteins and other molecules at primary amines (-NH_2_) to form stable dye–protein conjugates. Labeled protein was aliquoted at 2 mg/mL at −80 °C until use.

### 5.3. Cell Lines

The Madin–Darby canine kidney (MDCK) cell line was purchased from ATCC (CCL-34) and maintained in DMEM-F12 medium supplemented with L-glutamine, 15 mM HEPES, 10% fetal bovine serum (FBS, Gibco/Invitrogen, Grand Island, NY, USA), and 50 U/mL of penicillin/streptomycin.

Cells were grown at 37 °C in a humidified atmosphere of 5% CO_2_.

### 5.4. Animals

Male OF1 Swiss mice weighing 20 g were housed in standard conditions in temperature-controlled, pathogen-free rooms with free access to standard pelleted food and tap water. The experiments were performed according to European Union guidelines for animal experimentation, and protocols were approved by the Animal Experimentation Ethics Committee of the University of Barcelona (Ref. number: 384/17). 

To perform in vivo studies, the experiments were carried out in the Animal Research Facility of the University of Barcelona (Bellvitge Campus) in animal preparation rooms equipped with the apparatus necessary for isoflurane anesthesia. 

Mice were i.v. injected with GFP-Etx or the GFP-Etx-H106P, GFP-Etx-ΔV108-F135, or GFP-Etx-ΔS188-F196 mutants at a final concentration of 2.5 μg/g in PBS-1% BSA. Prior to i.v. injections, the mice were anesthetized with isoflurane until the end of the experiment.

Mice were sacrificed by decapitation 10 min after the injection and were routinely processed for histopathological analysis. To study the effects over a longer period of time, some mice were sacrificed 2 h after the injection. Each experimental procedure was performed three times in duplicate.

In all cases, tissues were fixed by immersion in paraformaldehyde (PFA) for 12 h, embedded in paraffin, cut into 5 µm sections, and mounted on SuperFrost^®^ Plus microscope slides (#631-0108, SuperFrost, VWR, West Chester, USA). Slides were stained with hematoxylin and eosin (HE) and were examined in a Leica DMD108 digital microimaging system.

To perform in vitro studies, a mouse was anesthetized with an intraperitoneal administration of ketamine (100 mg/kg) and xylazine (10 mg/kg) and then perfused by a gravity-fed system with 4% PFA via the vascular system for 20 min. Tissue was fixed by immersion in PFA for 12 h and embedded in OCT, cut into 10 µm sections with a cryostat, mounted on SuperFrost^®^ Plus microscope slides, and stored at −20 °C until use. Sheep, bovine, and caprine kidney and bladder samples were obtained from the Mercabarna slaughterhouse in Barcelona. Tissues were fixed by immersion in PFA, embedded in OCT, and cut into sections as described above until use.

### 5.5. In Vitro Binding of GFP-Etx Mutants to MDCK Cells and Tissues

Cells were grown to confluence on coverslips and fixed with 4% PFA for 12 min at RT. After three washes with PBS, the cells were blocked with PBS containing 20% normal goat serum (NGS) and 0.2% gelatin for 1 h at RT. Then, the cells were incubated for 30 min with 200 nM GFP, GFP-pEtx, or GFP-pEtx mutants in PBS containing 1% NGS and 0.2% gelatin for 1 h at RT. After three washes with PBS, nuclei were stained with TO-PRO-3 (1:1000 dilution, Molecular Probes, Invitrogen) for 7 min, washed again, and mounted with the aqueous mounting medium Fluoromount (Sigma, Madrid, Spain). 

To study the binding of GFP-Etx to different animal tissues, slides were blocked, incubated with GFP-pEtx, GFP-pEtx mutants, or GFP, and mounted as described previously. To verify that the binding of GFP-pEtx-ΔS188-F196 to the proximal tubules was specific, co-incubations with the unlabeled pEtx-ΔS188-F196 at a molar ratio of 20:1 with respect to GFP-pEtx-ΔS188-F196 were performed in slices of mouse kidney for 1 h at RT. After the co-incubations, slices were washed with PBS and mounted as described above.

To determine that the binding of GFP-pEtx-ΔS188-F196 was restricted to the proximal tubules, kidney sections were pre-incubated for 30 min at RT with biotinylated *Datura stramonium* agglutinin at 10 µg/mL (DSA; Sigma, Madrid, Spain) to detect the proximal tubules or with biotinylated *Arachis hypogaea* agglutinin at 50 µg/mL (PNA; Sigma) to detect the distal and collecting tubules [43,44]. After three washes with PBS, the sections were incubated with GFP-pEtx-ΔS188-F196 and streptavidin-Alexa-546 conjugate (Molecular Probes, 1:500 dilution) in PBS containing 1% bovine serum albumin (BSA) for 1 h at RT. After three washes with PBS, the nuclei were stained, and sections were mounted as described above.

Coverslips and slides were examined under a Leica TCS-SL spectral confocal microscope (CCiTUB, Bellvitge Campus Biology Unit, L’Hospitalet de Llobregat, Spain).

### 5.6. Cytotoxicity Assays in MDCK Cells

MDCK cells were grown in 96-well plates, and the cytotoxic activity of Etx and the mutants (with or without GFP) was determined using the MTS Assay Kit (G3581, Promega, Madison, WI, USA). Briefly, the assay determines the capacity of living cells to reduce the yellow tetrazolium salt into a colored formazan dye that is soluble in cell culture media. The formazan dye is quantified by measuring the absorbance at 490 nm. 

Cells were incubated with pEtx, Etx, or the Etx mutants (with or without GFP) at 50 nM for 1 h at 37 °C. Triplicates of each condition were performed. 

### 5.7. Luciferin-Luciferase ATP Detection Assays in MDCK Cells

MDCK cells were plated in black 96-well plates with a clear flat bottom and grown to confluence in 100 µL of media. ATP released from the cells after GFP-Etx exposure was measured using the luciferin-luciferase method [24,25]. Once the basal recording signal was stable, GFP-pEtx, GFP-Etx, or the GFP-Etx mutants were added to each well to obtain the desired final concentration of 50 nM. When the peak of bioluminescence returned to the basal level, Triton X-100 was added to evaluate the content of ATP still present in the cells. Each condition was run in triplicate in three independent experiments. 

### 5.8. Oligomer Complex Formation in MDCK Cells

To analyze the formation of GFP-Etx and GFP-Etx mutant protein complexes, MDCK cells were grown to confluence in 10 cm tissue culture dishes and incubated with GFP-pEtx, GFP-Etx, or the respective mutants at 50 nM for 30 min at 37 °C.

After incubation, the cells were washed thrice with PBS and then scraped off with 100 µL of ice-cold RIPA buffer (25 mM Tris-HCl, pH 7.4, 150 mM NaCl, 1% NP40, 0.1% SDS, and 1% sodium deoxycholate) supplemented with a protease inhibitor cocktail (1:100; #P8340, Sigma-Aldrich, Madrid, Spain). Harvested cells were sonicated on ice using a UP50H ultrasonic processor (Hielscher, Ultrasound Technology, Berlin, Germany) at an amplitude of 80% for 10 cycles of 0.5 s each. The cells were centrifuged at 20,000× *g* for 10 min, and the pellet was discarded. Supernatants were centrifuged at 100,000× *g* for 30 min, and the pellets were resuspended in 100 µL of RIPA buffer and quantified using the PierceTM BCA Protein Assay kit (Thermo Scientific). Forty µg of each sample were electrophoresed in a precast polyacrylamide gel (Mini-PROTEAN^®^ TGX TM; #456-9033, Bio-Rad), transferred, and analyzed by Western blot using the anti-GFP antibody as described previously. 

For the loading control, membranes were incubated with an anti-α-tubulin clone DM 1A antibody (1:2000 dilution, #T9026, Sigma-Aldrich) followed by rabbit anti-mouse immunoglobulins/HRP (1:5000 dilution, #P0161, Dako). Membranes were developed with the LuminataTM Crescendo Western HRP substrate (Millipore), and signals were detected using an Amersham Imager 600 (GE Healthcare Life Sciences).

### 5.9. Immunohistochemical Analysis of Mouse Samples

Endogenous peroxidase activity in the slices was first blocked with 10% methanol (*v/v*) and 2% H_2_O_2_ (*v/v*) in PBS for 30 min. The slices were pre-incubated for 1 h at RT with PBS containing 20% NGS (Gibco, Paisley, UK), 0.2% Triton, and 0.2% gelatin (Merck, Darmstadt, Germany). Then, the sections were incubated overnight at 4 °C with the rabbit polyclonal anti-pEtx antibody [34] diluted at 1:100 in PBS containing 1% NGS, 0.2% gelatin, and 0.2% Triton. The samples were washed three times with PBS and incubated with an anti-rabbit EnVision+ system-HRP labeled polymer (#k4001, Dako) for 30 min at RT. The samples were washed three times with PBS, and the sections were developed by a peroxidase reaction, which was performed in a solution containing 0.6 mg/mL of 3,3′-diaminobenzidine substrate (DAB; D-5637, Sigma-Aldrich, Saint Louis, MO, USA) and 0.5 µL/mL of H_2_O_2_ in PBS for 5 min, before being stopped with PBS. As a control, sections were treated identically but without incubation with the primary antibody. The sections were counterstained with hematoxylin, and the slides were then dehydrated and mounted with a DPX mounting medium. The slides were examined under a Leica DMD108 digital microimaging system.

### 5.10. Treatment with Pronase E, Detergents, N-glycosidase F, and Sodium Hydroxide (NaOH)

To assess the nature of the binding of Etx-ΔS188-F196, kidney sections were treated with pronase E, detergents, N-glycosidase F, or NaOH prior to incubation with the mutant.

Pronase E participates in the hydrolysis of peptide bonds internal to a sequence and sequentially eliminates amino acids from the N and C termini. To determine whether a protein was directly involved in the binding of the mutant, kidney sections were treated with 0.5 mg/mL of pronase E (Sigma) in 0.1 M PBS at 37 °C for 30 min. As a control, sections were treated only with the buffer in the same conditions [23]. 

To verify the possible participation of N-glycosidic or O-glycosidic residues in the binding of the mutant, N-glycosidase F (Roche) was used to hydrolyze N-glycan chains from glycoproteins [23], while beta-elimination treatment with 0.055 N NaOH was used to hydrolyze O-glycan chains, given that alkaline conditions are used to release O-GalNAc-linked oligosaccharides from glycoproteins [23,45]. Sections were pre-incubated with 100 U/mL of N-glycosidase F overnight at 37 °C in 20 mM PB, pH 8, containing 10 mM EDTA. To verify the participation of O-glycosidic residues, beta-elimination treatment with 0.055 N NaOH for 1 h in water at 37 °C was performed [23]. *Datura stramonium* agglutinin (DSA; Sigma) and *Arachis hypogaea* agglutinin (PNA; Sigma) were used as a positive control in the treatments for N-glycan and O-glycan elimination, respectively [23,46,47]. 

To assess the sensitivity of the mutant to detergents, kidney sections were incubated with 2% Triton X-100 (Sigma), 2% sodium deoxycholate (Sigma), or 2% sodium cholate (Sigma) at RT for 1 h [23]. 

After the treatments, the sections were washed three times with PBS at RT and incubated with GFP-Etx-ΔS188-F196 for 1 h at RT. The sections were washed and mounted as described before. In the case of NaOH treatment, the reaction was stopped by adding 0.055 N sulfuric acid for 30 s before the PBS washes. Depending on the treatment, controls were performed by omitting pronase E, N-glycosidase F, NaOH, or detergents from the incubation buffer. The autofluorescence produced by the effect of the treatments on the tissue was digitally eliminated using the Adobe Photoshop program.

### 5.11. Statistics

Results from the cytotoxicity assays were assessed with nonlinear regression analysis using one-way ANOVA with the Dunnett multiple comparison test. Data from the luciferin-luciferase ATP detection assays were assessed using the luminescence values from the area under the curve for GFP-pEtx, GFP-Etx, or the GFP-Etx mutants in relation to the luminescence values from the area under the curve for Triton X-100 in each case. All statistical analyses were performed using GraphPad Prism version 7.00 for Windows (GraphPad Software, La Jolla, CA, USA).

### 5.12. Molecular Graphics

Molecular graphics and analyses were performed with UCSF Chimera, developed by the Resource for Biocomputing, Visualization, and Informatics of the University of California, San Francisco, with support from NIH P41-GM103311.

## Figures and Tables

**Figure 1 toxins-14-00288-f001:**
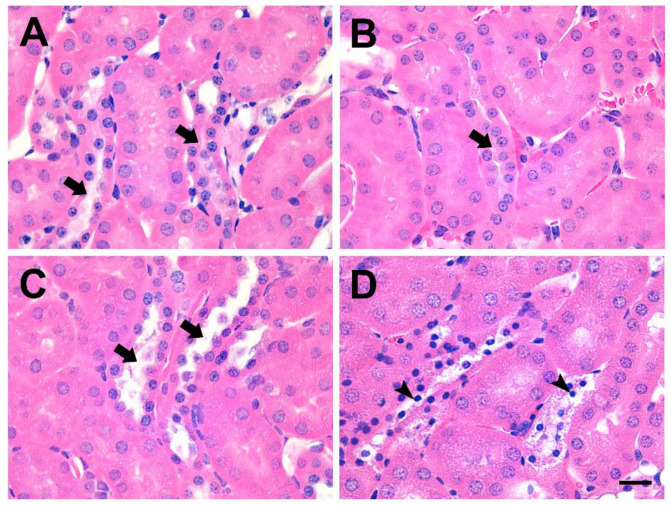
GFP-Etx mutants are non-toxic in the distal and collecting tubules of the kidneys after i.v. injection. HE staining of kidney sections from mice injected with GFP-Etx-H106P (**A**), GFP-Etx-ΔV108-F135 (**B**), GFP-Etx-ΔS188-F196 (**C**) or GFP-Etx (**D**). None of the mutants produced cytotoxic effects or pyknotic nuclei (arrows) in the epithelial cells from the distal and collecting tubules of the kidneys. This is in contrast to the cytotoxic effect and the pyknotic nuclei produced by GFP-Etx (arrowheads in (**D**)) in the epithelial cells. Note the condensed chromatin (deep blue nuclei, arrowheads in (**D**)) accompanied by cytoplasmic vacuolization of dead cells. Scale bar, 20 µm.

**Figure 2 toxins-14-00288-f002:**
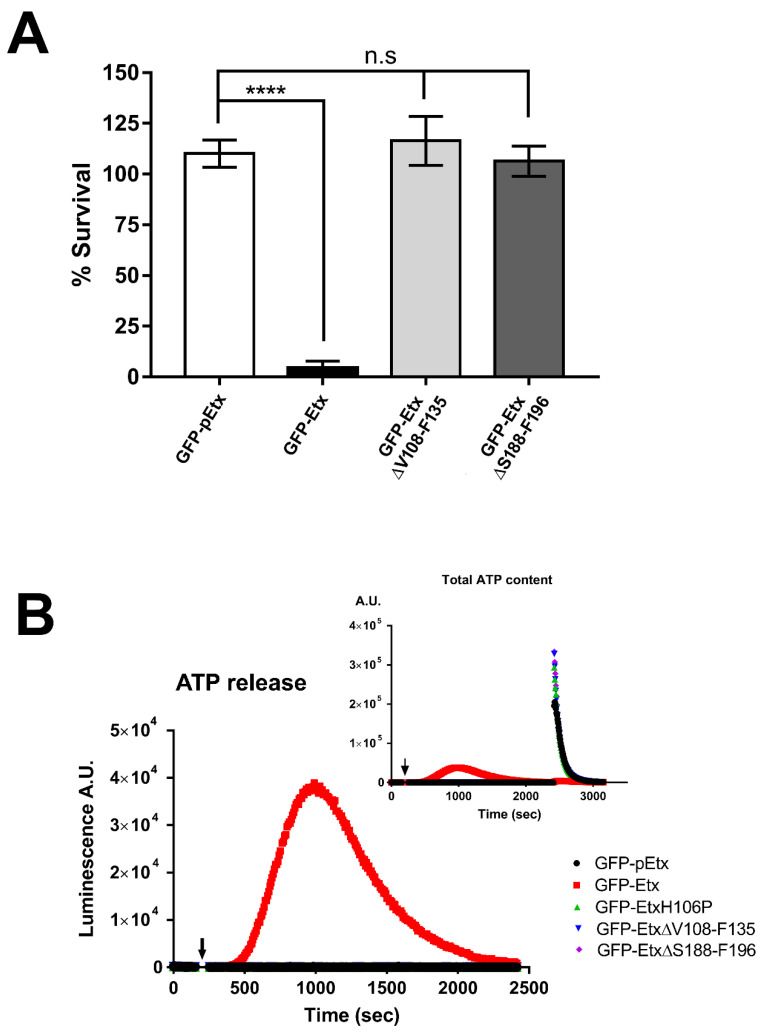
The mutants GFP-Etx-ΔV108-F135 and GFP-Etx-ΔS188-F196 are non-toxic in MDCK cells. (**A**) MTS assays revealed that the mutants GFP-Etx-ΔV108-F135 (light gray column) and GFP-Etx-ΔS188-F196 (dark gray column) were non-toxic in MDCK cells. GFP-pEtx (white column) was used as a negative control and GFP-Etx (black column) was used as a positive control. In all cases, the cells were incubated with 50 nM of the different forms of Etx at 37 °C for 1 h. (**** *p* < 0.0001), (ns = non-significant). (**B**) ATP release was detected by GFP-Etx action in MDCK cells (red line) but was not detected by GFP-pEtx (black line) GFP-Etx-H106P (green line), GFP-Etx-ΔV108-F135 (blue line), and GFP-Etx-ΔS188-F196 (purple line). Cells were treated with Triton X-100 for the last 15 min in order to quantify total ATP content (insert). Arrowhead marks the beginning of incubation with the toxins. All the toxins were incubated at 50 nM at 37 °C.

**Figure 3 toxins-14-00288-f003:**
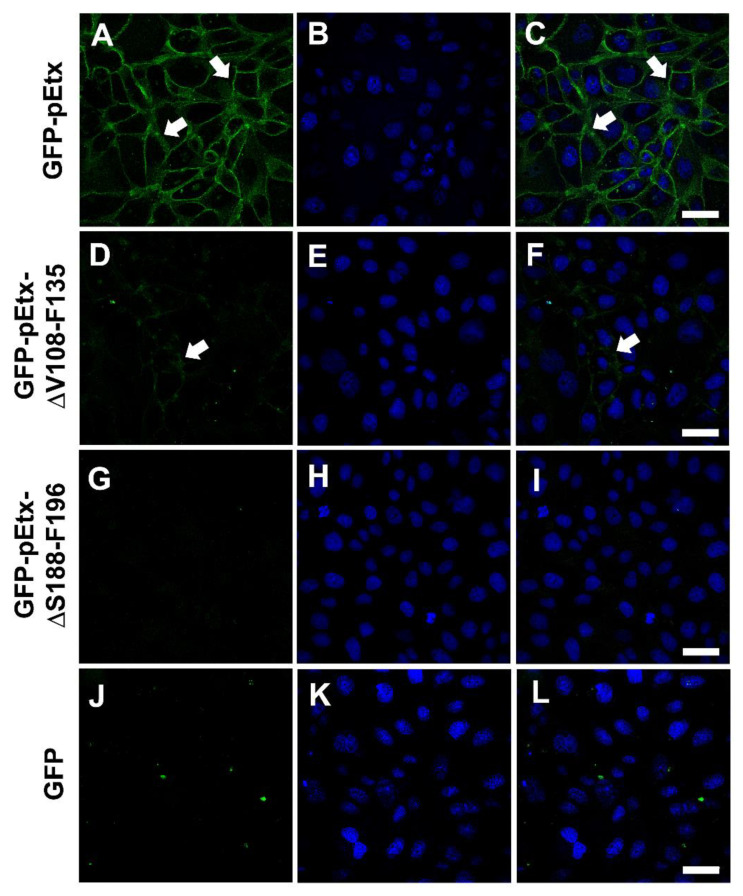
Strongly reduced binding of GFP-pEtx-ΔV108-F135 and no binding of GFP-pEtx-ΔS188-F196 to the plasma membrane of MDCK cells. Confocal microscopy images reveal a strong reduction in the binding of GFP-pEtx-ΔV108-F135 to the plasma membrane of MDCK cells (arrows, (**D**–**F**)) compared to the positive control GFP-pEtx (in green, arrows, (**A**–**C**)). No binding of GFP-pEtx-ΔS188-F196 to the plasma membrane was detected (**G**–**I**). GFP alone was used as a negative control (**J**–**L**). Nuclei were stained with TO-PRO-3 (blue). Scale bar, 40 µm.

**Figure 4 toxins-14-00288-f004:**
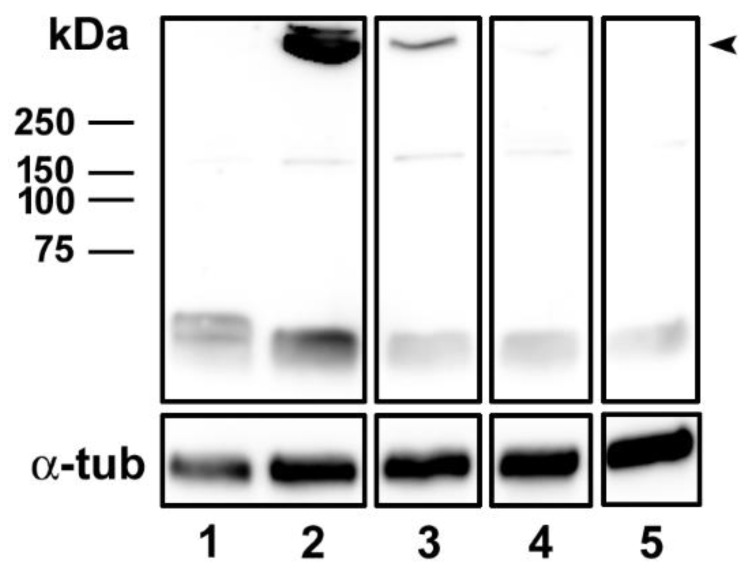
A strong reduction in the oligomerization of GFP-Etx-ΔV108-F135 and no oligomerization of GFP-Etx-ΔS188-F196 in MDCK cells. MDCK cells were treated with 50 nM of GFP-pEtx (lane 1), GFP-Etx (lane 2), GFP-Etx-H106P (lane 3), GFP-Etx-ΔV108-F135 (lane 4), or GFP-Etx-ΔS188-F196 (lane 5) for 30 min at 37 °C. Note the formation of an oligomer complex with a high molecular weight (above 250 kDa, black arrowhead) for GFP-Etx (lane 2). A thinner band was detected for GFP-Etx-H106P (lane 3), and a very faint band was observed for GFP-Etx-ΔV108-F135 (lane 4). No oligomer complex formation was detected for GFP-Etx-ΔS188-F196 (lane 5). α-tubulin was used as the loading control.

**Figure 5 toxins-14-00288-f005:**
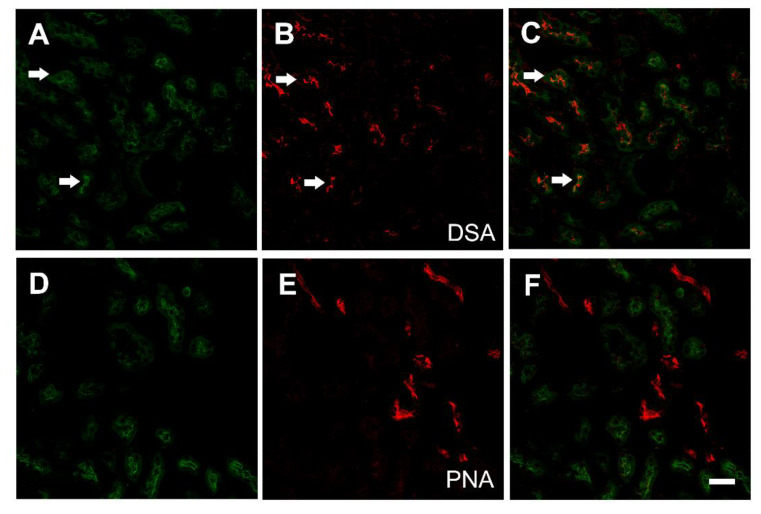
The mutant GFP-pEtx-ΔS188-F196 binds to the proximal tubules of the kidneys. Confocal images of mouse kidney co-incubated with GFP-pEtx-ΔS188-F196 (green, (**A**,**C**,**D**,**F**)) and the lectins DSA (red, (**B**,**C**)) or PNA (red, (**E**,**F**)). Note the colocalization of GFP-pEtx-ΔS188-F196 with the lectin DSA at the proximal tubules (arrows in (**C**)). No colocalization with the lectin PNA, which recognizes the distal tubule, was detected. Scale bar, 40 µm.

**Figure 6 toxins-14-00288-f006:**
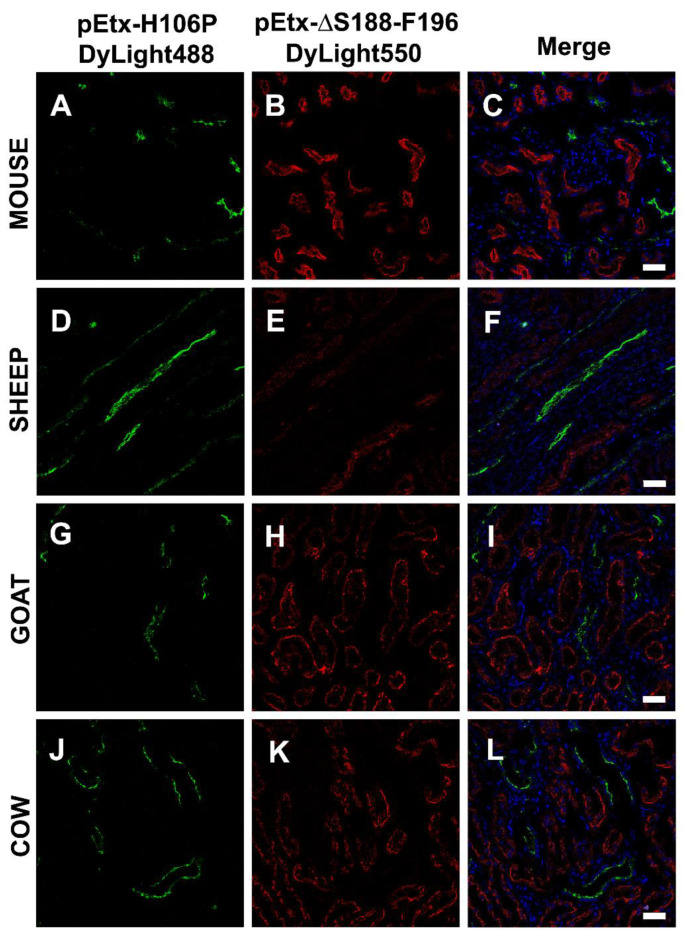
Co-staining of distal and proximal kidney tubules with the pEtx mutants: pEtx-H106P-DyLight488 and pEtx-ΔS188-F196-DyLight550. Confocal images of kidney sections from mouse (**A**–**C**), sheep (**D**–**F**), goat (**G**–**I**), and cow (**J**–**L**). The mutant pEtx-H106P was labeled with DyLight488 (green in A, D, G, and J), while the mutant pEtx-ΔS188-F196 was labeled with DyLight550 (red in B, E, H, and K). Note the binding of pEtx-ΔS188-F196 to the proximal tubules and the binding of pEtx-H106P to the distal tubules in all the species tested. Nuclei were stained with TO-PRO-3 (blue). Scale bar, 40 µm.

**Figure 7 toxins-14-00288-f007:**
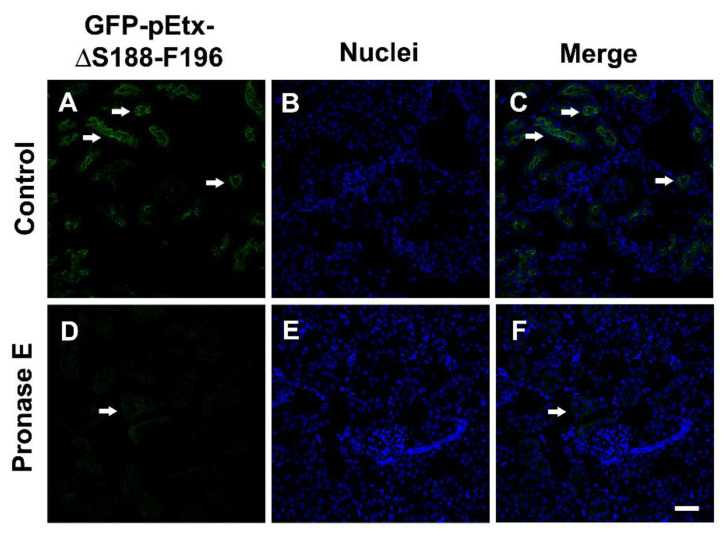
Effect of pronase E on the binding of GFP-pEtx-ΔS188-F196 to the proximal tubules of mouse kidneys. Mouse kidney sections were preincubated for 30 min at 37 °C with 0.5 mg/mL of pronase E (**D**–**F**) or without pronase E under the same conditions (**A**–**C**). The binding of GFP-pEtx-ΔS188-F196 to the proximal tubules was strongly reduced after pronase E treatment. Nuclei were stained with TO-PRO-3 (blue). Scale bar, 40 µm.

**Figure 8 toxins-14-00288-f008:**
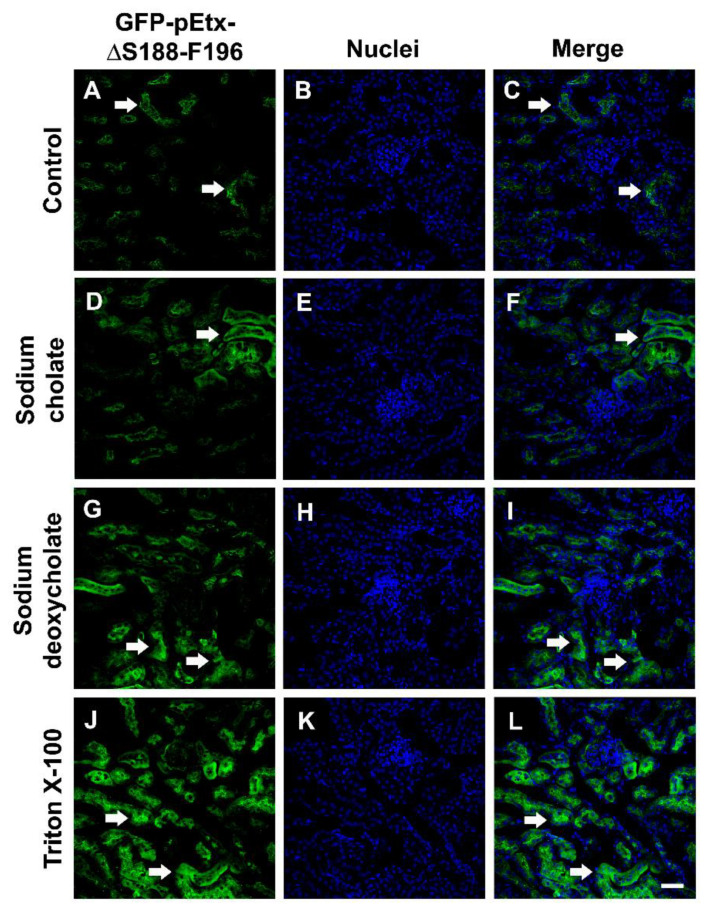
Binding of GFP-pEtx-ΔS188-F196 to the proximal tubules after detergent treatments. Mouse kidney sections were pretreated with 2% sodium cholate (**D**–**F**), 2% sodium deoxycholate (**G**–**I**), or 2% Triton X-100 (**J**–**L**) for 1 h at RT. As a control, sections were treated only with buffer under the same conditions (**A**–**C**). The binding of GFP-pEtx-ΔS188-F196 to the proximal tubules did not decrease after detergent treatment, and there was also a slight increase in the binding (arrows in (**D**,**F**,**G**,**I**,**J**,**L**)) compared to the control (arrows in (**A**,**C**)). Nuclei were stained with TO-PRO-3 (blue). Scale bar, 40 µm.

**Figure 9 toxins-14-00288-f009:**
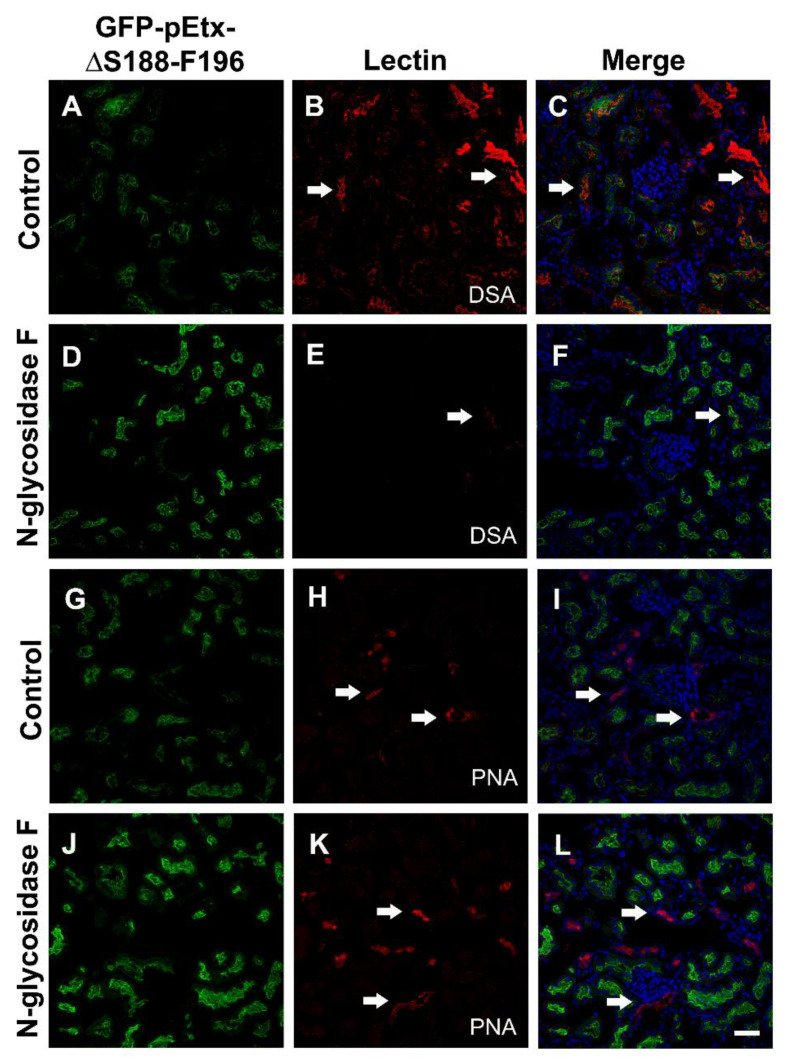
Effect of N-glycosidase F on the binding of GFP-pEtx-ΔS188-F196 to the proximal tubules. Confocal images of mouse kidney sections pre-treated with 100 U/mL of N-glycosidase F overnight at 37 °C (N-glycosidase F) or without the enzyme in the same conditions (control). A slight increase in the binding of GFP-pEtx-ΔS188-F196 to the proximal tubules was observed after treatment (compare (**D**,**J**) with (**A**,**G**)). Note a reduction in the binding of the *Datura straomium* lectin (DSA), which is used for N-glycan detection (compare arrows in (**E**) and (**B**)). No changes were observed in the binding of the *Arachis hypogaea* lectin (PNA), which recognizes O-glycans (compare arrows in (**K**) and (**H**)). Nuclei were stained with TO-PRO-3 (blue, **C**, **F**, **I**, **L**). Scale bar, 40 µm.

**Figure 10 toxins-14-00288-f010:**
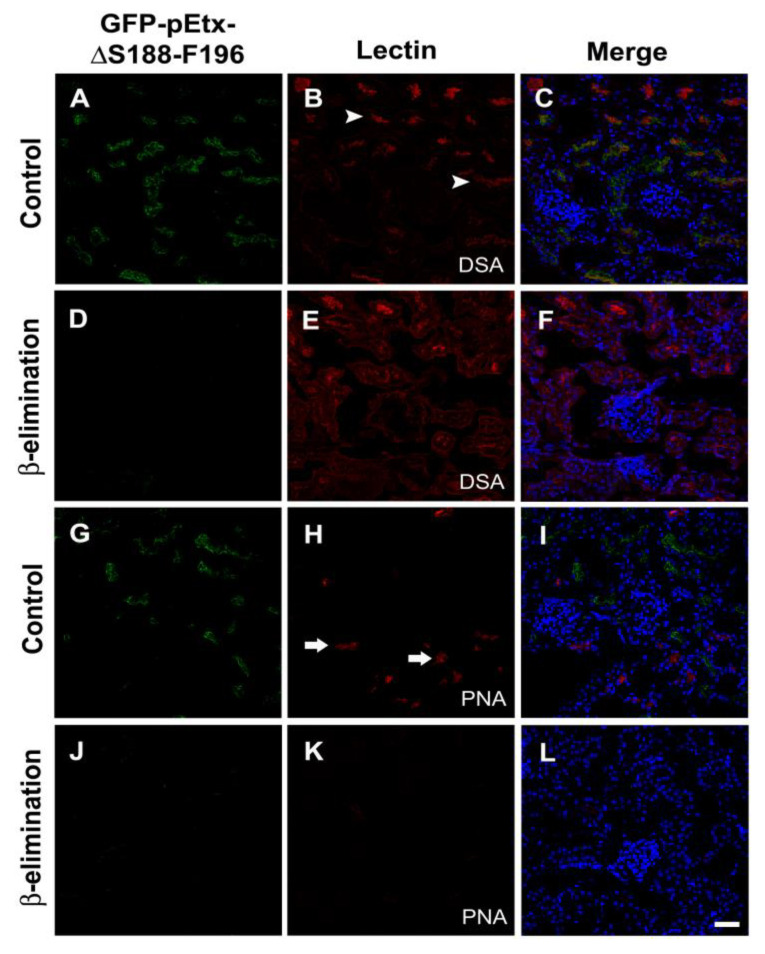
Effect of β-elimination on the binding of GFP-pEtx-ΔS188-F196 to the proximal tubules. Confocal images of mouse kidney sections pre-treated with 0.055 N NaOH for 1 h at 37 °C (β-elimination) or without NaOH in the same conditions (control). Sections were co-incubated with GFP-pEtx-ΔS188-F196 (green, (**A**,**C**,**D**,**F**,**G**,**I**,**J**,**L**)) and the lectins DSA (red, (**B**,**C**,**E**,**F**)) or PNA (red, (**H**,**I**,**K**,**L**)). Note the usual DSA staining of mainly the proximal tubules (arrowheads in (**B**)), which increased notably after treatment (red, (**E**)). The PNA lectin shows staining mainly of the distal tubules (arrows in (**H**)), which decreased notably after treatment (red, (**K**)). The binding of GFP-pEtx-ΔS188-F196 to the proximal tubules diminished after treatment (**D**,**F**,**J**,**L**) compared to the usual staining without NaOH treatment (**A**,**C**,**G**,**I**). Nuclei were stained with TO-PRO-3 (blue). Scale bar, 40 µm.

**Table 1 toxins-14-00288-t001:** Forward and reverse primers used with the QuickChange multi-site-directed mutagenesis kit.

Primers	Sequences
Forward pEtxΔV108-F135	5′ACTGATACAGTAACTGCAACTACTACTCATACTGCAAATACAAATACAAA3′
Reverse pEtxΔV108-F135	5′TTTGTATTTGTATTTGCAGTATGAGTAGTAGTTGCAGTTACTGTATCAGT3′
Forward pEtxΔS188-F196	5′GTAAAGTTAGTAGGACAAGTAAGTGGAAGTTTATCGGATACAGTAAAT3′
Reverse pEtxΔS188-F196	5′ATTTACTGTATCCGATAAACTTCCACTTACTTGTCCTACTAACTTTAC3′
Reverse Active wild-type Etx	5′TTTATTCCTGGTGCCTTAATAGAAA3′

## Data Availability

Not applicable.

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
