# Peer review of "New Mutants of Epsilon Toxin from Clostridium perfringens with an Altered Receptor-Binding Site and Cell-Type Specificity"

_toxins, 2022, doi:10.3390/toxins14040288_

Round 1

Reviewer 1 Report

Thank you for considering the requests and suggestions.

Author Response

We thank the reviewer for the comments and suggestions in the first round. As there are no comments and new suggestions in the revised version we hope that now the manuscript has been improved to be published in toxins.

Reviewer 2 Report

This reviewer would like to express congratulations to authors to complete the sophisticated work in this manuscript. Here only a few minor comments to improve the manuscript are listed.

  1. Figure 1 and 2: Though figure legends are shown, difference among A-D may not be clearly understood by readers, unless they are all specialists of pathology. Please try to make any modification of legends or presentation of figures.
  2. FigS5, S6, and S7 are included in main text. Overall, number of figures are too many. Consider to select essential ones in main text, and others to be move to supplementary material.
  3. In this manuscript, authors concluded functional mapping of Etx. Readers may be interested in potential of the development of vaccine from this study. Are the mutants prepared in this study immunogenic and is neutralizing antibody induced? Any relevance to vaccine development should be mentioned in Discussion, even briefly.

Author Response

Please, find attached the response to the reviewer.

Thanks for the comments and suggestions that have improved the manuscript.

This manuscript is a resubmission of an earlier submission. The following is a list of the peer review reports and author responses from that submission.

Round 1

Reviewer 1 Report

The manuscript "Differential binding site of the epsilon toxin mutant Etx ΔS188-F196 from Clostridium perfringens" describes results of a study on two mutant forms of epsilon toxin.

The paper is generally well written, experiments and results are described in a clear manner and reveal some interesting perspectives for research on epsilon toxin and pore-forming toxins in general. The manuscript is therefore of interest for the readership of Toxins and the scientific quality of the results is sufficient for publication in Toxins. I nevertheless request some additional experiments and a few minor changes.

My main requests are:

The results on both toxins are interesting. The study nicely combines in vitro experiments using MDCK cells and binding studies on tissues incubated with GFP-labelled toxin mutants.

The cell culture experiments are used to show oligomer formation and activity of the toxin mutants. Here I believe the authors must show that the lack of activity of the two new mutants on MDCK cells is not due to the addition of the bulky GFP molecule. Although the WT ETX-GFP serves as a valuable control, I would request that the authors also confirm lack of activity in the non-GFP-tagged versions of the two mutant toxins, which they actually show as being expressed in Figure S2.

Results of the binding studies clearly show a change in binding affinity to proximal tubular cells in the Etx ΔS188-F196 mutant. The authors conclude that the receptor binding is altered and that therefore the toxin does not bind to the original receptor and does not integrate into the membrane as a pore. This interesting "story" is however not completed in the manuscript and readers are left with an open question: is there an effect on proximal tubular epithelial cells, which the mutated form binds to? This question could be answered by additional in vitro experiments using cultured proximal tubular epithelial cells. Authors should attempt to show binding, oligiomer formation and activity on such cells. Such cells would at least be available as human proximal tubular epithelial cells and maybe these cells are also available from other species. The species independent binding of this etx form to cells on mouse, ovine, caprine and bovine tissues might allow using human cells cultures to test this hypothesis further. As mentioned above I would also request that both, the GST tagged and the untagged toxins are tested.

Minor changes:

The title only refers to one toxin mutant (the manuscript is on two) and additionally does not really reflect the most important take home message from the study. I believe the main conclusion is that the alteration of the two different domains alters receptor binding and cell type specificity of the toxin. Please reformulate the title.

Authors use a previously described inactive GFP-Etx-H106P mutant as control. This mutant and the rational for using it as a control should be shortly and properly introduced when the first results of experiments are shown/described (page 3, line 97 and Figure 1). This information is important for the readers to understand the experiments and only comes little by little in the subsequent text and figures (e.g. page 13, lines 346 ff).

Figure 3: Title states null binding of both toxin mutants, which I would interpret as no binding at all.  However next sentence it is stated (as shown in figure) that binding is strongly reduced in the ΔV108-F135 mutant. Please correct wording of title.

Page 12-13 first two paragraphs: This to me is more part of the introduction than the discussion and in fact it is partly redundant to information  given in the introduction. Consider shortening and moving information to introduction.

Conclusions: Last sentence: The new toxin mutant could be used to detect proximal tubule. Why would this be important? I think the importance of the findings is rather that mutations of the respective domain seems to alter receptor affinity and this could be used to further characterize the toxin-receptor interactions.

Reviewer 2 Report

The manuscript is dedicated to the study of Etx toxin from Clostridium perfringens. The authors made two mutants of this toxins with impaired domain 1 and 2. They have found, that mutant with deletion of the loop in the domain 1 not only lost toxicity, but also changed its specificity. Some comments:

  1. line 361: should be PSA instead of DSA.
  2. I would use fig 4s instead of fig 5 in the main text, because it is very difficult to distinguish proximal and distal tubules in fig 5.
  3. Fig 4: all the lanes should be loaded on one gel, not on several different, when quantity matters.
  4. I would add the info what Pronase E and beta-elimination does.
  5. If the mutant deltaS188-F196 has only domain 1 impaired, which is responsible for receptor binding, and this mutant specifically bound the proximal tubules epithelium, why it was unable to form pores and to kill cells?